# Evolution of the murine gut resistome following broad-spectrum antibiotic treatment

Laura de Nies[1,3], Susheel Bhanu Busi [1,3], Mina Tsenkova[2], Rashi Halder [1], Elisabeth Letellier [2✉] & Paul Wilmes [1,2✉]

The emergence and spread of antimicrobial resistance (AMR) represent an ever-growing healthcare challenge worldwide. Nevertheless, the mechanisms and timescales shaping this resistome remain elusive. Using an antibiotic cocktail administered to a murine model along with a longitudinal sampling strategy, we identify the mechanisms by which gut commensals acquire antimicrobial resistance genes (ARGs) after a single antibiotic course. While most of the resident bacterial populations are depleted due to the treatment, *Akkermansia muciniphila* and members of the Enterobacteriaceae, Enterococcaceae, and Lactobacillaceae families acquire resistance and remain recalcitrant. We identify specific genes conferring resistance against the antibiotics in the corresponding metagenome-assembled genomes (MAGs) and trace their origins within each genome. Here we show that, while mobile genetic elements (MGEs), including bacteriophages and plasmids, contribute to the spread of ARGs, integrons represent key factors mediating AMR in the antibiotic-treated mice. Our findings suggest that a single course of antibiotics alone may act as the selective sweep driving ARG acquisition and incidence in gut commensals over a single mammalian lifespan.

---

[1] Systems Ecology Group, Luxembourg Centre for Systems Biomedicine, University of Luxembourg, Esch-sur-Alzette, Luxembourg. [2] Department of Life Sciences and Medicine, Faculty of Science, Technology and Medicine, University of Luxembourg, Esch-sur-Alzette, Luxembourg. [3]These authors contributed equally: Laura de Nies, Susheel Bhanu Busi. ✉email: paul.wilmes@uni.lu; elisabeth.letellier@uni.lu

Prior to the advent of antibiotics, bacterial infections were the leading cause of disease and mortality in humans. Antibiotic usage is now commonplace in treating infections[1], as well as ensuring the safety of surgical procedures[2,3] and organ transplantation[4]. In addition, they are extensively used in animal husbandry[5] and also in animal models for studying the gut microbiome[6–11]. Concomitantly, their prevalence and administration are widespread in developing and developed countries alike[12,13], whereby antibiotics are easily accessible. However, many bacterial taxa have evolved antimicrobial resistance (AMR) to several classes of antibiotics, and multidrug-resistant bacteria have now emerged, preventing the comprehensive treatment of infections and resulting in a growing number of deaths[14]. Due to the overall rise in resistance, as well as a lack in the development of new antibiotics, AMR has emerged as a growing global threat to human health[15]. Therefore, a clear understanding of the selective sweeps (processes or mechanisms driving increased frequency of genetic variation) underlying the evolution, timescale, and transmission of ARGs is of crucial importance.

In general terms, bacteria can acquire and develop AMR through two distinct genetic mechanisms, either through the acquisition of spontaneous mutations during replication of the bacterial genome or through the accumulation and dissemination of resistance genes via mobile genetic elements (MGEs). MGEs, such as bacteriophages (phages), plasmids, and integrons, promote the transfer of resistance genes between bacterial populations through the process of horizontal gene transfer (HGT)[16]. Although the majority of bacteria are harmless commensals, which do not cause disease, they still provide a rich repertoire of resistance genes[17]. As such, through HGT, opportunistic pathogens may acquire resistance genes from other commensals. Therefore, MGEs conferring multidrug resistance are considered the cause for the wide dissemination of the now globally prevalent AMR "superbugs"[14]. Furthermore, while resistant bacteria may remain latent, they contribute to the overall reservoir of AMR based on which resistant pathogens may emerge once selective pressure due to antibiotics is built up[18,19]. Compounding this phenomenon, the overuse of antibiotics both in the treatment of human disease and in animal husbandry has fueled the build-up of AMR globally[14]. With the realization that the gut microbiome plays crucial role in disease etiology[20], antibiotic-treated animal models remain one of the methods by which the intestinal microbiome is studied[20,21]. The potential caveats, however, of using antibiotics for modulating the endogenous populations, including the emergence of AMR is unknown.

In this work, utilizing a mouse model, we assess the effect of selective antibiotic sweeps on the evolution of AMR within the commensal gut microbiome over a single mammalian lifespan after a single course of antibiotic treatment. We find that taxa such Akkermansia muciniphila and species within Enterobacteriaceae, Enterococcaceae, and Lactibacillaceae families are recalcitrant to antibiotic treatment. Moreover, we describe the key roles played by integrons in mediating antibiotic resistance against the administered antibiotic cocktail. Our observations allow us to test whether specific bacteria, including commensals, are more susceptible or capable of acquiring ARGs, as well as assess the influence of HGT on shaping the gut microbiome's resistome.

## Results

### Selection of specific taxa due to antibiotic-mediated depletion of the gut microbiome.
To assess the effect of selective pressures on AMR evolution, two groups of mice were single-housed in specific-pathogen-free conditions, whereby one of the groups was treated with an antibiotic cocktail (ampicillin, vancomycin, metronidazole, and neomycin) representing broad-spectrum antibiotic treatment regimens used in preoperative procedures and in animal models ("Methods"). Longitudinal fecal samples were collected from the control and antibiotic-treated mice prior to treatment, i.e., day 0, and subsequently immediately after treatment, i.e., day 7, and two weeks after treatment, i.e., day 21 (Fig. 1a).

Although there was a drop in weight during the antibiotic treatment phase, mice treated with antibiotics did not show any significant differences in weight compared to the control mice (Supplementary Fig. 1). We assessed the overall microbiome profiles at days 0, 7, and 21 and observed a major shift in the community profiles of the antibiotic-treated mice on days 7 and 21 (Supplementary Fig. 2). At the level of metagenome-assembled genomes (MAGs), taking only into account high-quality genomes (> 90% complete and < 5% contamination), we found a significant enrichment in Akkermansia muciniphila and Ligilactobacillus spp. on day 21 after antibiotic treatment (Two-way ANOVA, adjusted $p$-value < 0.05), despite a near-total depletion of the microbiota at day 7 in the treated mice (Fig. 1b). Simultaneously, several genera such as Alistipes, Odoribacter, members of Muribaculaceae, and Prevotella (including CAG-95, CAG-485, CAG-873) were significantly decreased or depleted entirely at days 7 and 21 (Two-way ANOVA, adjusted $p$-value < 0.05; Fig.1b), demonstrating the potency of the antibiotic cocktail. Concomitantly, the overall functional potential encoded by the metagenomes of the antibiotic-treated mice was shifted (PERMANOVA $p$ < 0.05; Supplementary Fig. 3a). More specifically, the functional complement, i.e., potential functions encoded by the metagenome, demonstrated a significant enrichment (Two-way ANOVA, adjusted $p$-value < 0.05) in pathways relating to signaling molecules (e.g., quorum sensing) on days 7 and 21 compared to the controls (Supplementary Fig. 3b). Interestingly, at day 7, we also observed a significant decrease in the abundance of genes involved in the biosynthesis of secondary metabolites in the antibiotic-treated mice, which may be associated with the reduction of the microbial community owing to the antibiotic treatment.

### Resistome after antibiotic treatment.
Due to the differences observed in the functional potential associated with interactions and secondary metabolites, we further assessed the abundance of ARGs in the control and treated mice using PathoFact[22]. We observed a significant increase in ARGs within the treated group (adjusted $p$-value < 0.05, Wilcoxon rank-sum tests; Fig. 2a). More specifically, the antibiotic-treated mice exhibited significantly higher abundances in ARGs compared to the control mice directly after antibiotic treatment on day 7. These levels were maintained during recovery until day 21. We further found that within the treated mice the overall ARG abundance was significantly higher after treatment at days 7 and 21, compared to day 0 (adjusted $p$-value < 0.05, Wilcoxon rank-sum tests). We found that the overall ARG abundance in the treated mice was also significantly enriched at days 7 and 21, when compared to the control mice at the same timepoints. On days 7 and 21, we specifically observed AMR against aminoglycoside, aminoglycoside:aminocoumarin, beta-lactam, fluoroquinolone, fosfomycin, glycopeptide, fusidic-acid, phenicol, mupirocin, triclosan and multidrug to be significantly enriched within the treated group (Fig. 2b). Of these, three resistance categories, namely aminoglycoside (neomycin), beta-lactam (ampicillin), and glycopeptide (vancomycin), could be directly linked to the administered antibiotics. While the other resistance categories are not associated with the administered antibiotics, it is likely that they were

a

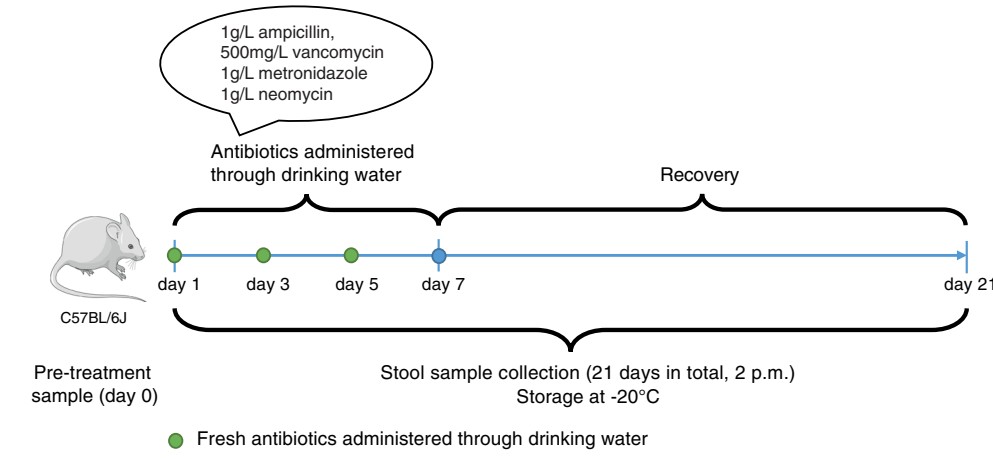

**Fig. 1 Experimental design and metagenome-assembled genome profiles. a** Representative illustration demonstrating the project overview, including dates and collections of treatment, and fecal sample collection. Eight single-housed mice per group were treated with an antibiotic cocktail (treatment) or water (control), longitudinally. Fecal samples collected at days 0, 7, and 21 were used for microbiome and antimicrobial resistance profiling. **b** Genus level representation of the MAGs recovered from control and treatment groups – pre and post (day 0, day 7, and day 21 respectively) antibiotic administration. Source data are provided as a Source Data file.

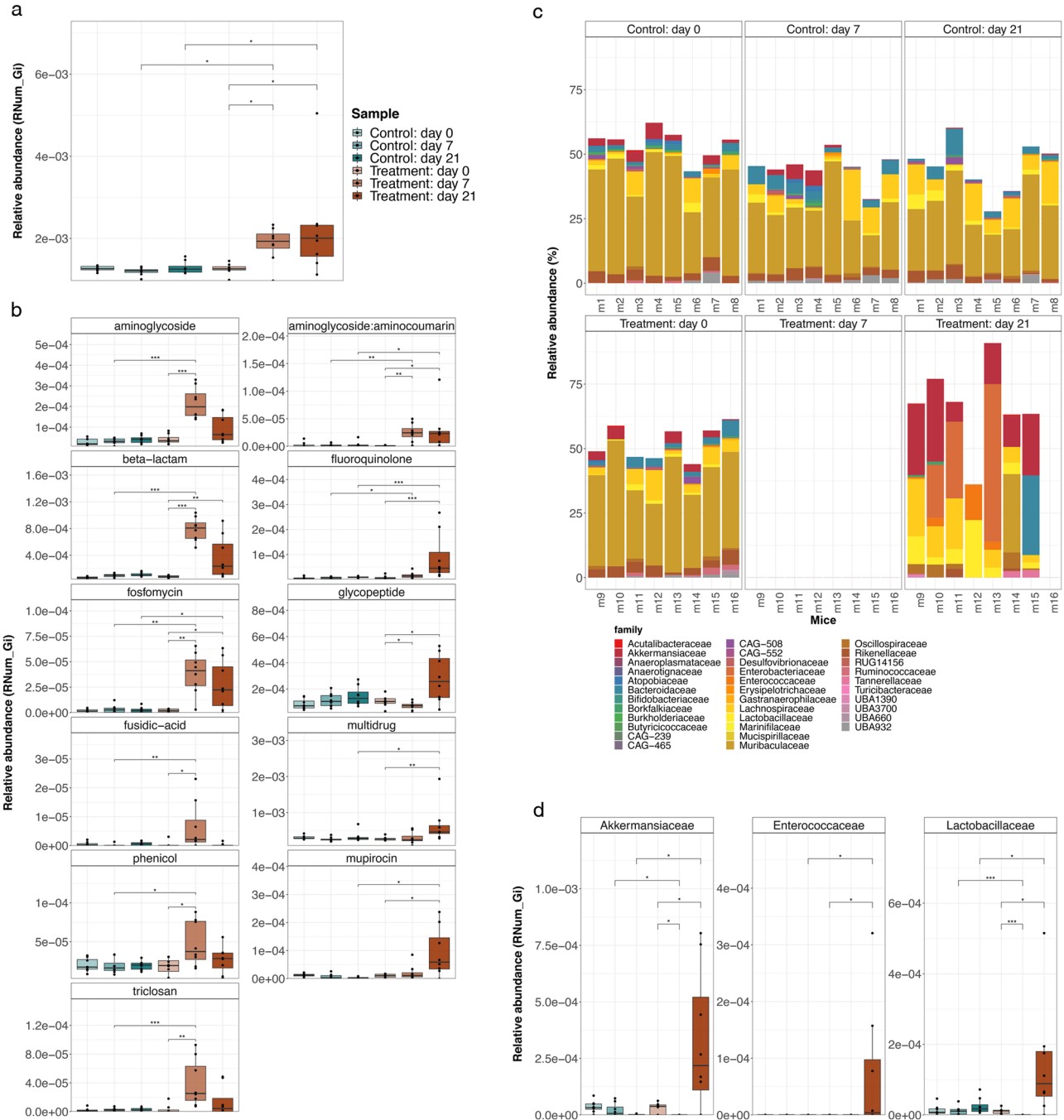

**Fig. 2 Resistome in antibiotic-treated mice. a** Overall ARG relative abundance levels are shown for each group ($n = 8$ biological replicates). *represents significance with an adjusted $p$-value less than 0.05 as assessed using a Wilcoxon rank-sum test. The center line denotes the median value (50th percentile), while the outer lines of the box represent the 25th to 75th percentiles. The black whiskers mark the 5th and 95th percentiles. **b** Significantly differentially abundant AMR categories found to be enriched in the mice treated with antibiotics compared across different timepoints, i.e., day 0, day 7, and day 21, *adjusted $p$-value < 0.05 (Wilcoxon rank-sum test). $n = 8$, biological replicates per group. The center line denotes the median value (50th percentile), while the outer lines of the box represent the 25th–75th percentiles. The black whiskers mark the 5th and 95th percentiles. **c** Barplots showing the relative abundance of MAGs (Family level) associated with ARGs in each sample. **d** Relative abundance of ARGs associated with Akkermansiaceae, Enterococcaceae, and Lactobacillaceae in the control and treated mice ($n = 8$ biological replicates per group). *adjusted $p$-value < 0.05 (Wilcoxon rank-sum test). The center line denotes the median value (50th percentile), while the outer lines of the box represent the 25th–75th percentiles. The black whiskers mark the 5th and 95th percentiles. Significance for all analyses was assessed using a two-sided Wilcoxon rank sum test, where, $p$-values are indicated by *, i.e., * < 0.05, ** < 0.01, *** < 0.001. Source data are provided as a Source Data file.

indirectly selected due to their co-localization, along with other resistance genes[23].

**Antibiotic-induced changes in taxonomic composition.** Since the metagenomes revealed an enrichment in different ARG categories, we investigated taxa harboring these ARGs. We linked ARGs to individual genomes by identifying contigs encoding ARGs and their corresponding assignment to MAGs including taxonomic classification of the MAGs using GTDBtk[24]. Based on the MAGs, we subsequently compared taxa contributing to AMR between the groups, including mice treated with antibiotics and those without. Interestingly, we did not recover any MAGs at day 7, likely due to the antibiotic-mediated depletion of the microbiota. However, in contrast to the MAGs, based on operational taxonomic units (OTUs), at day 7 we detected taxa that consisted predominantly of Bacteroidales spp. (Supplementary Fig. 2d). While taxa contributing to AMR within the control group remained constant, a shift in ARG-encoding taxa was observed within the treatment group after recovery, at day 21 (Fig. 2c). Alongside the increase in the abundance of several taxa (Fig. 2c), we found that the abundance of overall ARGs was increased in taxa belonging to the Akkermansiaceae, Enterococcaceae, and Lactobacillaceae families across all treated mice, as well as compared to the control group, at day 21 (Fig. 2d). Given the enrichment in ARGs at day 21, as opposed to days 0 and 7, and the specific lack of Akkermansiaceae, Enterococcaceae, and Lactobacillaceae MAGs at day 7, it is likely that the observed ARGs were acquired over time, rather than being encoded as intrinsic resistance mechanisms.

**MGEs linked to AMR dissemination.** MGEs are an established mechanism for the dissemination of AMR. To determine the function of MGEs in conferring the resolved ARGs under selective pressure, we analyzed the genomic context of the ARGs. The majority of the resistance genes were encoded on the bacterial chromosome (Fig. 3a). To assess the role of bacteriophages and plasmids in AMR transmission, we compared two strategies, where in the first approach, PathoFact was used to predict AMR on phages and plasmids obtained via metaviralSPAdes and metaplasmidSPAdes, respectively. In the second approach, PathoFact was used to identify both MGEs and ARGs from the metagenome assemblies obtained via IMP ("Methods"). We found that the standalone PathoFact detected more MGEs including those associated with AMR, compared to the SPAdes-based approach (Supplementary Fig. 3c). Furthermore, our analyses revealed that ARGs were encoded on both phages and plasmids (Fig. 3a). Interestingly, a reduction in the general abundance of ARGs mediated via plasmids was observed at day 7 in the treated mice compared to the controls, but the overall levels recovered by day 21. In contrast, phages linked to AMR were significantly enriched in the treated mice at day 7, compared to both pre-treatment at day 0 and the control mice at day 7 (Fig. 3a, b). Moreover, when analyzing the specific categories of the resistome, we found an increase in the abundance of phage sequences linked to aminoglycoside, aminoglycoside:aminocoumarin, and beta-lactam resistance, in conjunction with the administered antibiotic cocktail (Fig. 3c). Significant differences in these phage-associated AMR categories were observed in the treated group compared to the controls at day 7 and also within the group when comparing phage-associated ARG levels between days 0 and 7 (Fig. 3c).

To further investigate the effect of HGT on the evolution of AMR we identified and characterized all HGT events within the samples pre- (day 0) and post-treatment (days 7 and 21). Using MetaCHIP[25] and subsequent manual analyses ("Methods")

verifying the full-length matches for horizontally-transferred genes, we assessed whether ARGs were transferred across MAGs within the timepoints. We did not observe differences in the overall number of HGT events between the control and treatment groups at all timepoints (Fisher's exact test). However, to further assess whether HGT contributes to ARG spread, HGT events were specifically linked to AMR genes. We did not find a significant correlation between AMR and HGT in the control and treated mice across all timepoints, although *Akkermansia* spp. was involved in AMR-associated HGT events in 6/8 (i.e., 75%) of the mice treated with antibiotics (Supplementary Fig. 4a). Albeit not statistically significant (Fisher's Exact test, $p > 0.1$), *Akkermansia* was only involved in AMR-associated HGT in 3/8 (i.e., 37.5%) of the control mice (Supplementary Fig. 4b).

**Integrons mediate AMR in antibiotic-treated mice.** To further investigate the effect of antibiotic treatment on the evolution of AMR within the microbiota, we assessed the pangenomes of the significantly enriched and recalcitrant taxa in the treated mice, i.e., *Akkermansia muciniphila* and *Ligilactobacillus* spp. Interestingly, pangenome analyses of *Akkermansia muciniphila* revealed the acquisition of several genes at day 21 compared to day 0, including those mediated by integrases (Fig. 4a).

Horizontal gene transfer is typically attributed to phages and plasmids in metagenomes. However, integrons, often overlooked, play a key role in AMR dissemination and prevalence[26]. To evaluate the role of integrons in AMR, we assessed the abundance of *attC* sites and *intI* genes, both of which are required for efficient integron-mediated activity. We estimated the abundance of these genes on the same contig, including those that were associated with AMR categories. Overall, we found that ARGs abundant in antibiotic-treated mice were transferred via integrons (Fig. 4b and Supplementary Fig. 5a). Of these, there was a significant enrichment (adjusted *p-value* < 0.05, Wilcoxon rank-sum tests) in ARGs associated with complete (presence of *attC* and *intI* genes) integrons in mice at day 21 compared to day 0 and also when compared to the controls (Fig. 4b).

Additionally, these integron-mediated ARGs (complete, gene cassettes, and incomplete) were further analyzed to identify their putative genomic locations on phages or plasmids, since they are known to be carriers of integrons, thus elaborating on the method of integron-mediate AMR transmission. Interestingly, we identified several integron-mediated ARG cassettes encoded on plasmids at day 21 in the antibiotic-treated mice (Fig. 4c and Supplementary Fig. 5b).

As we identified antibiotic-induced changes of the microbial composition, we further investigated the association of AMR-encoding integrons with the microbial community. For this, we linked the AMR-associated 'complete' integrons with the reconstructed genomes and found that a substantial number was associated with genomes from families including Akkermansiaceae, Lachnospiraceae, and Enterobacteriaceae (Fig. 4c and Supplementary Fig. 5c). This finding reinforces our earlier findings with respect to enriched taxa and potential ARG-mediated mechanisms of resistance through integrases.

## Discussion

The classes and uses of antibiotics have been extensively developed since the fateful discovery of "mold juice" by Alexander Fleming[27,28]. Their use, and their overuse, has led to unrecoverable and irreversible states of resistance[29], resulting in an "arms race" where newer and more potent molecules[30] are becoming a necessity to ward off otherwise-susceptible bacteria. Even though antibiotics may result in the emergence of multi-resistant pathogens, their expanding use in medicine, especially

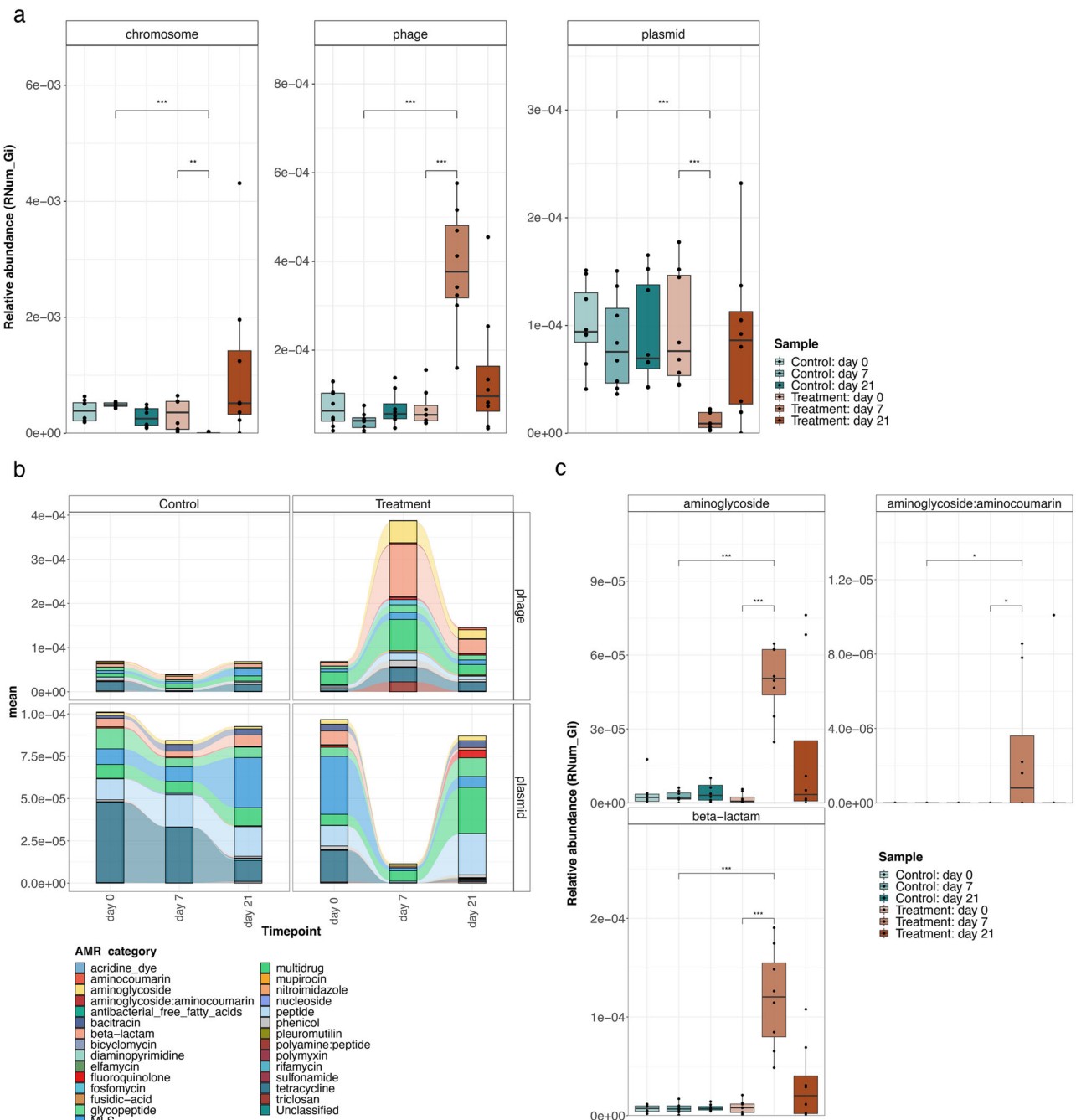

**Fig. 3 Abundance levels of resistome categories. a** Relative abundance of AMR encoded on the bacterial chromosome and those mediated via MGEs, such as bacteriophages (phages) and plasmids. $n = 8$ biological replicates per group. The center line denotes the median value (50th percentile), while the outer lines of the box represent the 25th–75th percentiles. The black whiskers mark the 5th and 95th percentiles. Significance was assessed using a two-sided Wilcoxon rank sum test, where, p-values are indicated by *, i.e., * < 0.05, ** < 0.01, *** < 0.001. **b** Abundance levels of AMR categories disseminated via phages and plasmids. categories pre- and post-treatment (day 0 and day 21 respectively). **c** Abundance levels of aminoglycoside, aminoglycoside:aminocoumarin, and beta-lactam resistance genes mediated via phages in the control and treated mice ($n = 8$ biological replicates). Significance was assessed using a two-sided Wilcoxon rank sum test, where, adjusted p-values are indicated by *, i.e., * < 0.05, ** < 0.01, *** < 0.001. The center line denotes the median value (50th percentile), while the outer lines of the box represent the 25th–75th percentiles. The black whiskers mark the 5th and 95th percentiles. Source data are provided as a Source Data file.

as a means of modulating the gut microbiome, remains unquestionable[31]. For example, they have also been proposed as prophylactics for treating cancers[8,32] and modulating the gut microbiota[10,21]. Antibiotics not targeting *Clostridioides difficile* infection are commonly used within the first weeks of a fecal microbiota transplantation (FMT) as a standard therapy[33]. Similarly, preoperative antibiotic prophylaxis in humans is a common practice and typically involves three antibiotics (cefazolin, vancomycin, and gentamicin)[34,35] These are usually administered individually while vancomycin in combination with other antibiotics (e.g., cefazolin) has been proposed for treatment of methicillin-resistant *Staphylococcus aureus*[36]. Here, we hypothesized that antibiotic treatment would lead to an evolution of AMR in the commensal gut microbiome

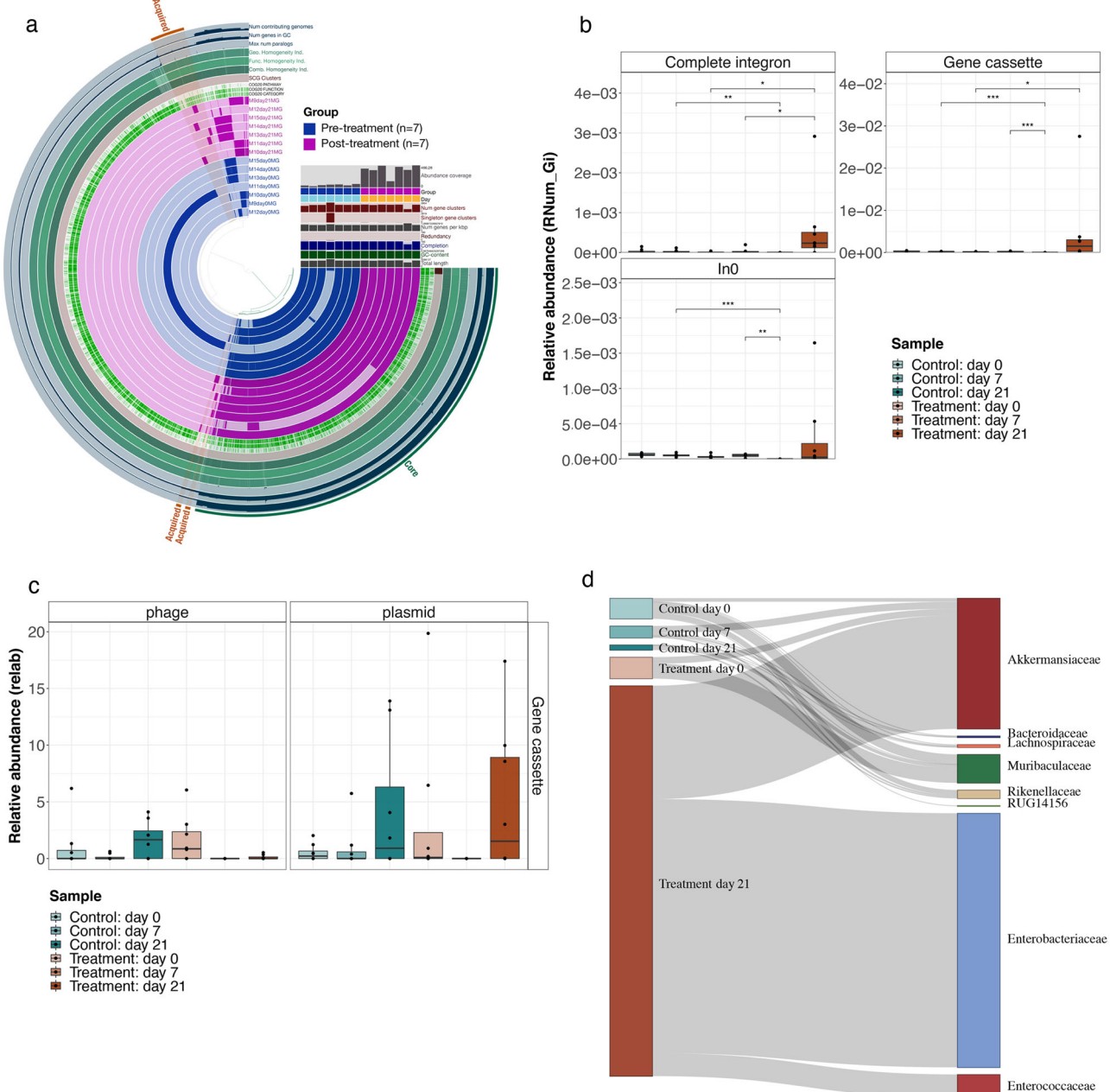

**Fig. 4 AMR-mediated via integrons in mice administered with antibiotics. a** Anvi'o based visualization of the *Akkermansia muciniphila* genomes from the pre- and post-treatment (day 0 and day 21 respectively) samples in blue and purple respectively. Annotations on the outer ring indicate the 'Core' genome and the 'Acquired' genome partially (75%) mediated by integrons. GC: gene clusters; Geo. Homogeneity Ind.: geometric homogeneity index; Func. Homogeneity Ind.: functional homogeneity index; Comb. Homogeneity Ind.: combined homogeneity index. **b** Boxplots showing the relative abundance of ARGs linked to integrons. $n = 8$ biological replicates per group. The center line denotes the median value (50th percentile), while the outer lines of the box represent the 25th–75th percentiles. The black whiskers mark the 5th and 95th percentiles. A two-sided Wilcoxon rank sum test was used to assess significance, where, p-values are indicated by *, i.e., * < 0.05, ** < 0.01, *** < 0.001. **c** Relative abundance of gene cassettes found to be encoded by phages and plasmids across different groups ($n = 8$ biological replicates) and timepoints. The center line denotes the median value (50th percentile), while the outer lines of the box represent the 25th–75th percentiles. The black whiskers mark the 5th and 95th percentiles. **d** Alluvial diagram demonstrating the abundance of 'complete' integrons linked with observed ARGs in specific taxonomic families. The flow (gray) bars indicate the number of AMR-linked integrons found in each group. Source data are provided as a Source Data file.

population within a single animal lifespan and tested our hypothesis in a wild-type mouse cohort.

The antibiotics (ampicillin, vancomycin, metronidazole, and neomycin) were chosen given their utility in several mouse models[37,38] and in varying combinations in line with some clinical procedures[39]. We observed that *Akkermansia muciniphila* was significantly enriched whilst most taxa were depleted in mice

post-treatment, which is in line with other reports[40–42] where vancomycin treatment alone led to propagation of *A. muciniphila* or its dominance in the resistant commensal population. The resistance of this taxon can specifically be attributed to the presence of β-lactamase and nitroimidazole resistance genes reported by van Passel et al.[43] These findings also agree with the report by Palleja et al. in which the authors found that species harboring

β-lactam resistance genes were positively selected during antibiotic exposure[44], which is likely the case in our study since we observed higher ARG abundances at day 21 and not prior to antibiotic treatment. Alternatively, the observed ARGs could also be due to the acquisition of resistance genes possibly via lateral or horizontal gene transfer as reported previously by Guo et al.[45].

In addition to an overall enrichment in A. muciniphila, we also observed an enrichment in the functional complement of A. muciniphila with respect to signaling molecules, specifically quorum sensing and cyclic dinucleotide signaling. Microbial communities are characterized by emergent properties that themselves are primarily shaped by microbial interactions[46]. These interactions include intra- and extracellular signaling, such as quorum-sensing (QS) and cyclic dinucleotide sensing, as a means of adapting to internal and external stimuli[47]. Due to the paucity of external stimuli from other bacteria following antibiotic treatment, it is plausible that QS functions were selected for as a means for recalcitrant community members to ramp up signaling functions to induce antibiotic tolerance. This is in line with reports of collective antibiotic tolerance[48], contributing towards AMR, which may be mediated via QS molecules leading to bacterial resistance gene expression in a density-dependent manner[48]. Furthermore, QS molecules have also been reported to regulate secondary metabolite synthesis[49]. However, we observed a depletion in genes involved in secondary metabolite synthesis following antibiotic treatment which is expected since a majority of the endogenous population is depleted. Alternatively, this phenomenon also suggests that selection for genes involved in signaling and secondary metabolite synthesis are somewhat uncoupled in our experimental murine model and, thus, are subject to different selective sweeps. Overall, our results highlight the role of key functions conferred by specific bacterial taxa in antibiotic-exposed communities and shed light on the shorter-term evolutionary processes shaping community assembly and composition.

Given the nature of the antibiotic cocktail treatment, we found several related ARGs in the metagenomes of the treated mice. More importantly, we observed significantly increased resistance at day 21, against three out of the four antibiotics used in our study protocol: aminoglycoside (neomycin), beta-lactam (ampicillin), and glycopeptide (vancomycin). We, however, did not recover any resistance genes against nitroimidazoles (metronidazole). Additionally, we found that several taxa in mice treated with antibiotics were directly linked to the resistance categories of the antibiotics that they were treated with. This suggests that the selective pressure of the administered antibiotics may lead to real-time evolution of AMR within the gut microbiome. This is in line with a recent report by Xu et al.[50] albeit in a different mouse model (Balb/c), who found that treatment with single antibiotics lead to an increased abundance of resistance genes. Intriguingly, they only noted considerable levels of MGE-mediated horizontal gene transfer for fosfomycin but did not observe high levels of ARGs associated with integrases. In our study, we found that while phages and plasmids contributed to both HGT and AMR, integrons were also a key factor in AMR dissemination. Despite the decrease in overall diversity and number of taxa in antibiotic-treated mice, we observed a significantly increased prevalence of 'complete integron'-mediated AMR. In accordance with previous reports[51], we found that gene cassettes encoding ARGs are localized and mediated via MGEs, specifically plasmids. This has further implications since mobility allows penetration and potential integration of AMR into new taxa. Importantly, in conjunction with our findings about the survival and resistance of A. muciniphila, we found that several integrons associated with AMR were directly linked to this taxon.

Concerted knowledge has concentrated on pathogenic bacteria, and lately the emergence of AMR. Although the genome of A. muciniphila is thought to be plastic[45], our analyses emphasize the role of integrons in mediating AMR in commensals, including within this taxon. In addition, it is presently unclear whether the acquired ARGs are expressed, requiring the need for both experimental and cross-validation using methods such as meta-transcriptomics to resolve gene expression. Given the association of A. muciniphila with Parkinson's and other chronic diseases, our findings highlight the need to understand the role of integrons in mediating AMR within and beyond this taxon. Taken together, our data show that AMR is relevant to studies involving antibiotic treatment, especially within the commensal gut microbiome population. Our study is built on a systemic, and longitudinal design to understand the stage at which the resistance genes are acquired following antibiotic treatment. Overall, we highlight the need for understanding the real-time evolution of AMR in microbiome research, including functional and evolutionary consequences of integron-mediated AMR.

## Methods

**Ethical considerations.** The animal experiment was performed according to all applicable laws and regulations, after receiving approval by the institution's Animal Experimentation Ethics Committee at UL (AEEC) and the Ministry of Agriculture, Viniculture and Rural Development (LUPA 2019/99). The care and use of animals for research purposes was conducted according to the EU Directive 2010/63/EU, as well as the Grand-Ducal Regulation of 11 January 2013 on the protection of animals used for scientific purposes. These included the justification of the use of animals, their welfare and the incorporation of the principles of the "3Rs" (Replacement, Reduction and Refinement).

**Power calculation and sample size estimation.** To determine the number of animals required per treatment and control group we performed a multifactorial power analysis based on a 2015 study by Raymond et al.[52]. For this, we estimated the Jensen–Shannon Divergences (JSD) of the microbial profiles of the antibiotic-treated (cefprozil) and control groups. Based on the observed JSD, the inter-group variability was significantly high, thereby highlighting a minimum sample size of three mice per group to attain a power of 80 and a 5% alpha error rate, reflecting changes in microbial composition (Supplementary Fig. 6).

**Mouse model and antibiotic exposure.** Eight-week old C57BL/6J mice were purchased from Charles River Laboratories and experiments were performed according to all applicable laws and the regulations described under Ethical considerations. To limit individual variation of the gut microbiome in experimental groups, mice of the same age (8 weeks) were obtained from the same vendor and the same location in the vendor facility. After a 7-day quarantine and subsequent acclimation period of one week, mice were maintained in single housing conditions. Mice were housed in Allentown NexGen Mouse 500 (194 mm × 130 mm × 381 mm) cages (Allentown, USA) with JRS Rehofix Corncob bedding. Mice had access to reverse osmosis water with 2ppm of chlorine fed ad libitum along with standard A40 chow diet (SAFE, France). The animals were maintained under standard habitat conditions (humidity: 40–70%, temperature: 22 °C) with a 12:12 light cycle. Two groups of mice were established (control and treatment), and each group contained 8 animals (4 males + 4 females). Antibiotics, ampicillin (1 g/L), vancomycin (500 mg/L), metronidazole (1 g/L), and neomycin (1 g/L) were chosen for their utility in several mouse models[37,38] and in line with some preoperative procedures[39]. They were administered as a cocktail within the drinking water to the treatment group starting at 8 weeks of age. Antibiotics were administered during a period of one week, after which the change was made to regular drinking water for the duration of the recovery period. Fecal samples were collected prior to treatment and subsequently daily for a duration of 19 days (both treatment and recovery phase) starting prior to the antibiotic treatment till take down (Fig. 1a).

**Fecal processing and nucleic acid extraction.** A total of 48 fecal samples were obtained across three timepoints, i.e., prior to treatment: day 0, immediately after treatment: day 7, and after recovery: day 21, from each of the mice. 50 mg of frozen stool samples were aseptically weighed into sterile vials. Genomic DNA was isolated with the DNeasy PowerSoil Kit (Qiagen, USA) including an additional incubation step at 65 °C and milling, as described previously[53]. A minimum of 200 ng of total DNA was recovered from all the samples, yielding sufficient DNA for metagenomic sequencing including high-resolution, artefact-curated metagenomic data for subsequent analyses[54]. DNA extracted from all timepoints was thereafter stored at −80 °C until further use.

**DNA sequencing.** All DNA samples were subjected to random shotgun sequencing. Briefly, 200 ng of DNA was used for metagenomic library preparation using the Westburg NGS DNA Library Prep Kit (cat. no. WB 9096, Westburg Life

Sciences, Netherlands). The genomic DNA was enzymatically fragmented for 12 min and DNA libraries were prepared without PCR amplification. The average insert size of libraries was 480 bp. Prepared libraries were quantified using Qubit (ThermoFisher Scientific, USA) and quality checked on a Bioanalyzer instrument (Agilent, USA). Sequencing was performed at the LCSB sequencing platform (RRID SCR_021931) on a NextSeq500 (Illumina, USA) instrument using 2×150bp read lengths.

**Data processing for metagenomics, including genome reconstruction.** The Integrated Meta-omic Pipeline (IMP; v3 - commitID #6f1badf7)[55] was used to process paired forward and reverse reads using the built-in metagenomic workflow as previously described[56]. The workflow includes pre-processing, assembly, genome reconstruction, and functional annotation of genes based on custom databases in a reproducible manner. After trimming the adapters, the preprocessing step included the removal of *Mus musculus* (GRCm38.p6 (GCA_000001635.8); retrieved on 16-May-2020 from https://www.ensembl.org/Mus_musculus/Info/Index) reads. Thereafter the de novo assembly was performed using the MEGAHIT (version 2.0) assembler[57]. Default IMP parameters were retained for all samples. Metagenomic operational taxonomic unit (mOTU) profiles were generated from the trimmed and preprocessed reads to generate microbiome profiles for the control and treatment groups using mOTUs v2.5.1[58]. Concurrently, we used MetaBAT2[59] and MaxBin2[60] for binning in addition to an in-house binning methodology previously described[56] for genome reconstructions, i.e., MAGs. Subsequently, we obtained a non-redundant set of MAGs using DASTool v1.1.4[61] with a score threshold of 0.7 for downstream analyses, and those with a minimum completion of 90% and less than 5% contamination as assessed by CheckM v1.1.3[62]. Taxonomy was assigned to the MAGs using the extensive database packaged with gtdbtk v1.7.0[24]. To generate pangenomes, we collected all the bins taxonomically identified as *Akkermansia muciniphila* and used the anvi'o-based pangenome workflow described by Eren et al. (http://merenlab.org/2016/11/08/pangenomics-v2/)[63]. One of the treated mice (#16), was excluded from the pangenome analyses due to the unavailability of MAGs.

**Identification of antimicrobial resistance genes and association with mobile genetic elements.** We used PathoFact v1.0, a pipeline for the prediction of virulence factors and AMR genes, to predict and identify ARGs within our metagenomes. The assembly files from individual samples were used as input for the AMR analyses. To assess the relevance of metaplasmidSPAdes and metaviralSPAdes for identifying plasmid and viral sequences respectively, we used de novo SPAdes assembler v3.15.4[64]. For the assembly, we used the same kmer settings as the assembly setup in IMP, i.e., 21, 33, 55, and 77, in a paired-end format with 24 threads. Subsequently, ARGs were collapsed into categories based on the Comprehensive Antibiotic Resistance Database (CARD)[65] and identified using PathoFact. Thereafter, the relative abundance of the ARGs was calculated using the Rnum_Gi method described by Hu et al.[66].

Identified ARGs and their categories were linked to associated bacterial taxonomy using the metagenomic bin classifications. Furthermore, utilizing PathoFact, ARGs were linked to predicted mobile genetic elements (MGEs: phages and plasmids) to identify probable transmission of AMR between taxa. More specifically, to link both the MGEs and the taxonomy to the ARGs, we mapped the genes to assembled contigs, followed by identifying the corresponding bins (MAGs) to which the contigs belonged. By considering all different predictions of MGEs, a final classification was made based on the genomic contexts of the ARGs encoded on plasmids, phages, or chromosomes, including the classification of those that could not be resolved (ambiguous). The ARGs that could not be assigned to either the MGEs or bacterial chromosomes were further referred to as unclassified genomic elements. Certain ARGs were encoded on both the bacterial chromosome and phage genomes. HGT of ARGs was assessed using MetaCHIP v1.0[25] with a modified setting of full-length match of genes of interest to ensure robustness of the findings. The confirmation of ARGs and their associated mode of transfer was also performed manually alongside this by mapping identical 1Kbp flanking regions, via the same pipeline. Briefly, groups of genes among all input MAGs with maximum average identity were considered putative HGT genes. To validate the predicted candidates, a pairwise BLASTN was used to assess each pair of flanking regions of 10 Kbp. Visual representations of the genomic regions were extracted alongside the results for visual interpretation and inspection.

**Linking antimicrobial resistance genes with integrons.** The assemblies generated via IMP were used to assess the presence and abundance of integrons within the metagenomes. Briefly, *attC* sites were identified by HattCI[67] while for the annotation of the *intI* sites a BLAST database was created using the *intI* variant sequences from the UniProt database[68]. Only those contigs where both the signature genetic regions (*intI* and *attC*) were found were annotated as having 'complete' integron elements. We also identified the MAGs along with which the integrons were binned, thus linking the integrons to the reconstructed genomes. The ARG information was overlaid onto this to identify contigs where integrons were linked with ARGs. Furthermore, we used sequence coordinates to identify integron localization, i.e., chromosome, plasmid, or phage localization of gene cassettes, incomplete and complete integrons on MGEs. This information was used for downstream differential analyses.

**Data analysis.** Figures for the study including visualizations derived from the taxonomic and functional, were created using version 3.6 of the R statistical software package. GraphPad[69] was used to generate the figures for describing the longitudinal weight measurements of the mice. DESeq2[26] and Wilcoxon rank-sum tests with FDR-adjustments for multiple testing were used to assess significant differences for the AMR and taxonomic analyses whereas a paired two-way ANOVA (Analysis of Variance) within the *nlme* package was used for identifying statistically significant differences in the integron profiles. Chord diagrams for the HGT events were generated using scripts found within the MetaCHIP package[25] while the pangenome visualizations were obtained using anvi'o[63].

**Reporting summary.** Further information on research design is available in the Nature Research Reporting Summary linked to this article.

## Code availability

The open-source tools and algorithms used for the data analyses are reported in the Methods section, including relevant flags used for the various tools. The scripts and analysis codes are provided at https://git-r3lab.uni.lu/susheel.busi/intonate.

## Data availability

The sequencing data generated for this study are available via NCBI's SRA under the accession number: PRJNA691897. The metadata file indicating group and timepoint information can be obtained via the same accession ID. Raw counts for taxon abundances and genes generated in this study have been provided in the Source Data file. Source data are provided with this paper.

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

## Acknowledgements

Data analysis was performed using the HPC facilities[63] of the University of Luxembourg. L.d.N, M.T., and P.W. were supported by the Luxembourg National Research Fund (FNR) under PRIDE/11823097. S.B.B was supported by the Sinergia grant (CRSII5_180241) through the Swiss National Science Foundation to P.W in collaboration with Dr. Tom Battin at École Polytechnique Fédérale de Lausanne. This project has received funding from the European Research Council (ERC) under the European Union's Horizon 2020 research and innovation program (grant agreement No. 863664). This work was additionally supported by the Fondation du Pélican de Mie and Pierre Hippert-Faber under the aegis of the Fondation de Luxembourg (E-AGR-0023-10-Z; M.T.), the Action LIONS Vaincre le Cancer (E.L.), an Internal Research Project at the University of Luxembourg (R-AGR-0755-12; E.L. and P.W.) and funding from Club 51 Éislek to E.L. E.L. is also supported by the FNR and the Fondation Cancer Luxembourg under CORE/C20/BM/14591557. The mouse image was obtained under a CC-BY 3.0 license (https://creativecommons.org/licenses/by/3.0/) from smart.servier.com at www.bioicons.com.

## Author contributions

S.B.B., E.L., and P.W. conceptualized and designed the study. M.T. performed the animal experiments including housing, antibiotic treatment, and fecal sample collection. L.d.N. and S.B.B. did the DNA extractions, analyzed the data, and created the figures. R.H. and the Sequencing Platform sequenced the samples. All authors contributed to the preparation of the manuscript.

## Competing interests

The authors declare no competing interests.
