## [Peer Review File · Nature Communications]

Reviewers' Comments:

Reviewer #1:

None

Reviewer #2:

Remarks to the Author:

This review is limited to my area of expertise, i.e. bioinformatics (metagenomics) and statistical analysis.

1) The complete preprocessing pipeline (from raw reads to MAGs and annotation) should be available (e.g. github) for reproducibility. The IMP documentation is rather poor (<https://r3lab.uni.lu/web/imp/doc.html>) and last update dates back to 4 years ago (<https://git-r3lab.uni.lu/IMP/IMP>).

2) How did you predict mobile genetic elements (e.g. phages and plasmids) and how did you classify them? My opinion is that pipelines such as metaplasmidSPAdes and metaviralSPAdes (coupled with viralVerify) should be used in such cases in order to obtain the best results. Then, a taxonomical classification using plasmid and virus-curated databases should be performed.

3) The complete AMR pipeline (including integrons analysis) should be made available.

4) The R code (Data analysis section) should be made available.

5) Line 346: "This study does not use custom code. The open-source tools and algorithms used for the data analyses are reported in the Methods section, including relevant flags used for the various tools.": Also a bash and R scripts are "custom code" (see above).

Reviewer #3:

Remarks to the Author:

The authors determine the impact of a single oral antibiotic course of 4 antibiotics on mouse microbiomes, resistomes, and HGT resulting from treatment versus control. Antimicrobial resistance of the gut microbiome is exceptionally important, and the authors add to this growing body of literature, which commensals species (*Akkermansia* spp. mainly) are most enriched in resistance genes after treatment and that integrons appear to contribute to horizontal gene transfer. The findings are interesting and novel, but several major flaws call into question the conclusions. These major concerns are enumerated below along with several minor comments.

Major Comments:

- 1) The authors refer to the model as “over a single generation”. This is unclear as bacterial generations are on much shorter timescales. Similarly, the experiment lasts only 28 days, and no mouse offspring are analyzed (which would appropriately invoke the generational terminology). Perhaps the better terminology would be after a single antibiotic course. This would also be consistent with human literature (e.g., Palleja et al. 2018 *Nature Microbiology*) illustrating the effects on the gut microbiome of a single course of antibiotics. Further, since the experiment consists of one biological replicate with 4 technical replicates in each group, the reproducibility of the phenomena observed is unclear. The experiment needs to be repeated at least once more for this reviewer to have more confidence in the conclusions of the study.
- 2) The analysis and description for Figure 2D seems incomplete. The authors report a consistent increase in certain AMR genes within families. However, an average relative abundance is displayed so the reader cannot assess “consistent”. Furthermore, there is no description in the text or the figure for how these AMR genes are linked to these specific families. The figure also appears to show increased overall abundance of AMR genes in control mice at day 0. Perhaps the authors mean that certain AMR are increased in abundance and these correspond to AMR genes commonly found in *Akkermansiaceae* and *Lachnospiraceae*. It is not possible to make this conclusion from the figure, however. The relevant bars that are statistically significant should also have asterisks as it is unclear which comparisons are being made and which AMR genes are statistically significantly different between groups.
- 3) Figure 2D and Figure 3A are identical. Resolution of this could potentially solve problem 2 above, but this reviewer would need to see the appropriate figures.
- 4) Overall, the results section needs more methodological specifics. Although listed in the methods, it is helpful to have a brief phrase with each result of how that result was generated. For example, the authors “investigate the effect of HGT on the evolution of AMR.” It is unclear by what methodology and how the authors derive this conclusion.
- 5) Many figures are discussed as significantly different, but no depiction of which comparisons are statistically evaluated appears on the figure. It is therefore very challenging to interpret these figures (e.g., Supplementary Figure 4, Fig. 2D, 3A, etc.).

6) In determining the importance of integrons in HGT, the authors conclude that “AMR genes enriched in antibiotic-treated mice were transferred via integrons...” but there is no statistical support for this assertion.

Minor comments

Line 24: citation after husbandry appears to say “42”. Unsure what the actual reference is.

Line 29: multidrug-resistant bacteria are not “untreatable” just harder to treat with fewer antimicrobial agents. Phage therapy also useful for certain mycobacteria and Acinetobacter. Please revise

Line 45: “superbugs” should probably be in quotations.

Line 46 and 49 and 54-55 (and likely elsewhere): “antimicrobial resistance” should be AMR as this acronym has been introduced.

Supplementary Figure 1: Were any of the weight timepoints statistically different? If not, perhaps state “a non-significant drop during the antibiotic treatment phase”

Supplementary Figure 2A is hard to interpret because multiple levels of taxonomy are displayed with a count that sums to 1. It is unclear what is being plotted since the count should come to 100 for each taxonomic level separately. Plotting all mice at the same taxonomic level (genus or family) would aid in the interpretation of this figure.

Line 80: What does “With respect to the functional complement” mean? Please revise.

Figure 2D: this figure makes reference to phage and plasmid-based resistance genes but does not comment on this in the results main text. Please add a description

Line 114-115: The sentence beginning “subsequently, we did...” is incomplete. Please revise.

Supplementary Figure 4b-c: From the figure it is unclear how the authors arrive at the conclusion of an absence of “a significant correlation between AMR and HGT” as what is graphed (in b) is stacked barplots of AMR categories and in C is Akkermansia MAG with genes from other genera.

Supplementary figure 4D is never referenced in the results section.

Line 121-122: the sentence beginning with “However” requires a reference.

Figure 4: The legend text below the top level (group, num gene clusters) is blurry and hard to read.

Line 156-157 and line 233-234 is a misrepresentation of the reference. Ampicillin, vancomycin, metronidazole, and neomycin are not used together as surgical prophylaxis. Therefore, the statement “in line with most preoperative procedures” is not true. Surgical prophylaxis is most often accomplished with a single dose of cefazolin (Clinical Practice Guidelines for

Antimicrobial Prophylaxis in Surgery). Indeed, there are few recommendations for multiple simultaneous agents; one of which being for complicated intra-abdominal procedures contaminated with fecal material. Please revise.

Response letter to the reviewers' comments

The detailed authors' responses to the reviewers' comments are listed in blue text below with the line numbers where modifications were made being highlighted in yellow. Additionally, the authors numbered the comments (e.g. R1.1 for comment no.1 from reviewer 1) to improve readability.

Reviewer #1

Comment R1.1

"Reviewer #1 only provided comments to the editors questioning the strength of the claims. "

While reviewer #1 only provided comments to the editorial team, to improve the strength of our claims we have now included additional mice amounting to a total of 8 biological replicates per group. Moreover, we have included an interim timepoint, i.e. day 7, in addition to days 0 and 21 described in the original manuscript, to track the acquisition of antimicrobial resistance in a longitudinal manner. Therefore, the revised analyses includes a total of 48 samples, which underwent metagenomic sequencing and which we have further emphasized within the respective passages in the revised manuscript (lines 80-86, 313-330 and 346-349). We have also updated all figures with the additional timepoint and their respective samples.

Reviewer #2

Comment R2.1

The complete preprocessing pipeline (from raw reads to MAGs and annotation) should be available (e.g. github) for reproducibility. The IMP documentation is rather poor (<https://r3lab.uni.lu/web/imp/doc.html>) and last update dates back to 4 years ago (<https://git-r3lab.uni.lu/IMP/IMP>).

We recognize and appreciate the reviewer's concern regarding the documentation and updates on the mentioned IMP version (<https://git-r3lab.uni.lu/IMP/IMP>). Although the above-mentioned IMP version has indeed not been updated lately, a new version is now available at the following repository (<https://git-r3lab.uni.lu/IMP/imp3>; version 3; channel: IMPiris; commitD #6f1badf7) and was used for the present study. This has been highlighted within the Methods section (lines 454-456) of the revised manuscript.

Comment R2.2

How did you predict mobile genetic elements (e.g phages and plasmids) and how did you classify them? My opinion is that pipelines such as metaplasmidSPAdes and metaviralSPAdes (coupled with viralVerify) should be used in such cases in order to obtain the best results. Then, a taxonomical classification using plasmid and virus-curated databases should be performed.

We thank the reviewer for highlighting this point and for sharing their expertise regarding mobile genetic element (MGE) predictions. We used the PathoFact pipeline (PMID: 33597026) for the prediction of AMR genes. This also allows contextualization of the AMR genes with respect to MGEs including both on plasmids and phages. This has now been described in the Methods (lines 392-398) with reference to the PathoFact publication (PMID: 33597026). We have also clarified this in the Results section in lines 112-114 and 149-157.

With respect to the reviewer's comment, we would additionally like to highlight that there is no gold-standard approach for plasmid prediction. For our study, we decided to use PlasFlow (PMID: 29346586), which is included in our PathoFact pipeline, because it has been designed to classify contigs of already assembled metagenomic samples using a k-mer-based machine learning approach and does not rely on any marker genes making it more suitable for identification of known and novel plasmids. In contrast, the tool suggested by the reviewer, namely metaplasmidSPAdes, implements an assembly- and marker gene-based approach. While this strategy has its benefits, it also has several important limitations, originally highlighted in the corresponding publication (PMID: 27466620). Specifically, metaplasmidSPAdes relies on contig coverage differences and thus may fail if these are not present in the data. Moreover, metaplasmidSPAdes does not identify linear plasmids, whilst also misclassifying short chromosomal edges as plasmidic (false positives) and uses a predefined set of marker genes for final plasmid prediction which can also fail to identify plasmids (false negatives). Similarly, for the identification of viral sequences PathoFact uses DeepVirFinder (<https://doi.org/10.1007/s40484-019-0187-4>) which uses an assembly- and reference-free machine learning approach as compared to metaviralSPAdes.

Nevertheless, in accordance with the reviewer's suggestion we have assembled the reads for both plasmids and viruses using the 'SPAdes' workflow, subsequently annotating the AMR genes on these assemblies. Our results demonstrate that PathoFact performs better in terms of total number of contigs recovered wherein AMR genes were identified, including unique AMR genes. This information has been highlighted in the text in lines 152-159 and 392-396, and added as an additional figure (Supplementary Fig. 3c)

Comment R2.3

The complete AMR pipeline (including integrons analysis) should be made available.

We thank the reviewer for this suggestion. We have included a detailed description of the Methods including tools, scripts and commands for perusal and reproducible analyses via an open-access

page at: <https://git-r3lab.uni.lu/susheel.busi/intonate>. This information has also been highlighted in the revised manuscript in lines 454-456.

Comment R2.4

The R code (Data analysis section) should be made available.

All relevant code including for R analyses is now available at: <https://git-r3lab.uni.lu/susheel.busi/intonate>.

Comment R2.5

Line 346: "This study does not use custom code. The open-source tools and algorithms used for the data analyses are reported in the Methods section, including relevant flags used for the various tools.": Also a bash and R scripts are "custom code" (see above).

Please see the above responses to comments R2.3 and R2.4.

Reviewer #3

The authors determine the impact of a single oral antibiotic course of 4 antibiotics on mouse microbiomes, resistomes, and HGT resulting from treatment versus control. Antimicrobial resistance of the gut microbiome is exceptionally important, and the authors add to this growing body of literature, which commensals species (*Akkermansia* spp. mainly) are most enriched in resistance genes after treatment and that integrons appear to contribute to horizontal gene transfer. The findings are interesting and novel, but several major flaws call into question the conclusions. These major concerns are enumerated below along with several minor comments.

We thank the reviewer for their acknowledgement of the importance of our work in relation to our understanding of antimicrobial resistance within the gut microbiome especially given the novel findings with respect to specific commensals and the mechanisms underlying horizontal gene transfer. We greatly appreciate the reviewers' concerns and comments and have addressed these in detail in the responses to the comments below.

Major Comments:**Comment R3.1**

The authors refer to the model as "over a single generation". This is unclear as bacterial generations are on much shorter timescales. Similarly, the experiment lasts only 21 days, and no mouse offspring are analyzed (which would appropriately invoke the generational terminology). Perhaps the better terminology would be after a single antibiotic course. This would also be consistent with human literature (e.g., *Palleja et al. 2018 Nature Microbiology*)

illustrating the effects on the gut microbiome of a single course of antibiotics. Further, since the experiment consists of one biological replicate with 4 technical replicates in each group, the reproducibility of the phenomena observed is unclear. The experiment needs to be repeated at least once more for this reviewer to have more confidence in the conclusions of the study.

We appreciate the reviewer's concerns regarding the terminology of "over a single generation". In the original manuscript we referred to this term for "over a single generation of mice". We do however acknowledge that this may have been confusing. In this context, we have revised our title to "Evolution of the gut resistome following a selective antibiotic sweep". Additionally, we recognize that "after a single antibiotic course", as per the reviewer's suggestion, may indeed be appropriate and have adjusted the manuscript accordingly.

We would like to underline that our initial sample size was determined based on a multifactorial power analysis that was performed based on a 2015 study by Raymond *et. al* (PMID: 26359913). Specifically, we estimated the Jensen-Shannon Divergences (JSD) of the microbial profiles among those that were treated with cefprozil and compared them with those that received placebo treatments. Based on the observed JSD the inter-group variability was significantly high, demonstrating a minimum sample size of three per group (antibiotic-treated vs. control) to observe changes in microbial composition. Given the fact that the diversity of human microbial communities are much higher than that of mice (PMID: 25561744) and given the inclusion of longitudinal data, we believe that with an increase in sample size to 8 mice per group the power in our study exceeds the minimum number of 3 biological replicates required per group as shown by the power analyses. Details on the power calculation are now included in the updated manuscript including an additional figure (Supplementary Fig. 6). We have also updated the Methods to indicate this analysis in lines 313-320.

Regarding the experimental setup we want to highlight that each group *originally* consisted of 4 (single housed) biological replicates, instead of 4 technical replicates as mentioned by the reviewer. Nevertheless, to further address the reviewer's comment, and strengthen the study, we have now included additional data bringing the total number of "single-housed" mice to 8 animals per group. We also have added the microbiome and AMR profile data from an interim timepoint, i.e. day 7, after antibiotic treatment. This is described in further detail in the manuscript in lines 80-86 and 313-344.

Comment R3.2

The analysis and description for Figure 2D seems incomplete. The authors report a consistent increase in certain AMR genes within families. However, an average relative abundance is displayed so the reader cannot assess "consistent". Furthermore, there is no description in the text or the figure for how these AMR genes are linked to these specific families. The figure also appears to show increased overall abundance of AMR genes in control mice at day 0. Perhaps the authors mean that certain AMR are increased in abundance and these correspond to AMR genes commonly found in Akkermansiaceae and Lachnospiraceae. It is not possible to make

this conclusion from the figure, however. The relevant bars that are statistically significant should also have asterisks as it is unclear which comparisons are being made and which AMR genes are statistically significantly different between groups.

We thank the reviewer for these observations, comments and suggestions. As highlighted by the reviewer, Figure 2d previously represented an average relative abundance to indicate the different AMR categories transferred via plasmids and/or phages. In response to the reviewer's comment, we have now included in Figure 2c the contribution of various taxa to the overall AMR profiles. Furthermore, we have highlighted the significant differences with respect taxa mediating AMR in the revised manuscript in Figure 2d, and the corresponding level of statistical significance is indicated by asterisks (*). As per the reviewer's suggestion, we have edited the text in lines 88-104 and 140-143 and we have additionally revised and updated Figure 2 along with all the other figures to clarify the main messages.

As described in the Methods section (lines 403-414), AMR genes were linked to associated bacterial taxonomy using the metagenomic bin classifications. More specifically, AMR genes were mapped to assembled contigs, followed by identifying the corresponding bins to which the contigs belong. The corresponding text has now been updated in the revised manuscript in lines 134-136 to clarify how the AMR genes were linked to the corresponding taxa.

With respect to the reviewer's comment concerning apparent overall abundance of AMR genes in control mice at day 0, we did not observe any significant differences between the overall AMR abundances within the control mice, compared to the treatment group which in particular exhibited a significant increase in AMR genes post-treatment at days 7 and 21. Furthermore, as highlighted by the reviewer, our findings indicate that there is an increase in AMR gene abundances across several taxa. Of these, we observed a significant increase in AMR abundance linked to Akkermansiaceae, Lactobacillaceae and Enterococcaceae. We thank the reviewer again for their feedback and have clarified this further within the manuscript in lines 136-144.

Comment R3.3

Figure 2D and Figure 3A are identical. Resolution of this could potentially solve problem 2 above, but this reviewer would need to see the appropriate figures.

All figures have been updated with the additional data. The resolution of all figures has been set at 300 dpi.

Comment R3.4

Overall, the results section needs more methodological specifics. Although listed in the methods, it is helpful to have a brief phrase with each result of how that result was generated. For example, the authors "investigate the effect of HGT on the evolution of AMR." It is unclear by what methodology and how the authors derive this conclusion.

As mentioned by the reviewer, all methodologies used in this study are described in detail in the Methods section in lines 313-446. The Results section now focuses primarily on the relevant findings from our analyses. In accordance with the reviewer's suggestion, this has now been updated with brief details about the methods wherever appropriate. We have also updated the results section in lines 172-178, to describe that MetaCHIP and subsequent manual confirmation was used to identify full-length gene transfer matches across MAGs for AMR-encoding genes.

Comment R3.5

Many figures are discussed as significantly different, but no depiction of which comparisons are statistically evaluated appears on the figure. It is therefore very challenging to interpret these figures (e.g., Supplementary Figure 4, Fig. 2 D,3A, etc.).

We thank the reviewer for highlighting this important point. All the analyses where statistically significant differences were observed have been indicated as such in the updated figures and clearly highlighted in the text.

Comment R3.6

In determining the importance of integrons in HGT, the authors conclude that "AMR genes enriched in antibiotic-treated mice were transferred via integrons..." but there is no statistical support for this assertion.

We assessed the relative abundance of AMR genes encoded on integrons, plasmids and phages (Supplementary Fig. 5a), where we did not find significant differences between the three despite an increase in integron-encoded AMR genes at day 21. On the other hand, we found that complete integrons and the relative abundance of AMR genes encoded via these integrons were significantly enriched at day 21 compared to days 0 and 7 in the antibiotic-treated mice. However, this was not observed in the control group.

As described in lines 199-203, AMR genes statistically found to be increased in abundance after antibiotic treatment were contextualized regarding their localization on integrons. Subsequently, these integron-mediated AMR genes (complete, gene cassettes and incomplete (In0)) were further analyzed to identify their putative genomic localization on plasmids and phages, since plasmids are known to be carriers of integrons, thus elaborating on the method of integron-mediated AMR transmission. Per the reviewer's comment we have highlighted this further in lines 194-210 in the revised manuscript.

Minor comments

Line 24: citation after husbandry appears to say "42". Unsure what the actual reference is.

We thank the reviewer for picking up on this omission. Reference 42 referred to the publication He, Y et al. Antibiotic resistance genes from livestock waste: occurrence, dissemination, and treatment. *npj Clean Water* 3, 4 (2020). This has been updated in the revised manuscript.

Line 29: multidrug-resistant bacteria are not “untreatable” just harder to treat with fewer antimicrobial agents. Phage therapy also useful for certain mycobacteria and Acinetobacter. Please revise

We thank the reviewer for their suggestion. The sentence has been revised as suggested by the reviewer.

Line 45: “superbugs” should probably be in quotations.

We thank the reviewer for their suggestion. The passage has been revised as suggested by the reviewer.

Line 46 and 49 and 54-55 (and likely elsewhere): “antimicrobial resistance” should be AMR as this acronym has been introduced.

We thank the reviewer for their suggestion. The passages have been revised as suggested by the reviewer.

Supplementary Figure 1: Were any of the weight timepoints statistically different? If not, perhaps state “a non-significant drop during the antibiotic treatment phase”

We thank the reviewer for their suggestion and we have revised the passage accordingly.

Supplementary Figure 2A is hard to interpret because multiple levels of taxonomy are displayed with a count that sums to 1. It is unclear what is being plotted since the count should sum to 100 for each taxonomic level separately. Plotting all mice at the same taxonomic level (genus or family) would aid in the interpretation of this figure.

We thank the reviewer for their comment. In Supplementary Figure 2a, the y-axis represents the relative abundance scaled from 0 to 1, which represents the overall abundance summing to 100%. All the taxa are plotted at the genus level. Based on the metagenomic operational taxonomic unit (mOTU) database, the family name is indicated when the highest resolution for certain mOTUs is at the family-level. Taxonomic lineage resolution is contingent on the highest possible lineage identified for each mOTU. Therefore, per sequence-based taxonomy nomenclature, this is the appropriate representation of mOTUs belonging to a certain family. We have updated the figure

legend (lines 518-519) to indicate that all mOTUs are plotted at the Genus level, as suggested by the reviewer.

Line 80: What does “With respect to the functional complement” mean? Please revise.

We thank the reviewer for their suggestion. We have revised this section and we have added additional descriptions to the text in line 103.

Figure 2D: this figure makes reference to phage and plasmid-based resistance genes but does not comment on this in the results main text. Please add a description

We thank the reviewer for their suggestion. The figure has been updated and the appropriate description has been added in lines 140-143.

Line 114-115: The sentence beginning “subsequently, we did...” is incomplete. Please revise.

We thank the reviewer for picking this up. The passage has been modified.

Supplementary Figure 4b-c: From the figure it is unclear how the authors arrive at the conclusion of an absence of “a significant correlation between AMR and HGT” as what is graphed (in b) is stacked barplots of AMR categories and in C is Akkermansia MAG with genes from other genera.

To arrive at this finding the number of HGT events associated with each treatment group and timepoint was assessed using a Fischer’s Exact test. Based on the p -value being higher than the alpha error rate, we concluded as described. This information has now been updated in the revised manuscript in lines 174-184.

Supplementary figure 4D is never referenced in the results section.

We thank the reviewer for picking this up. This figure has been revised and does not include a ‘d’ panel.

Line 121-122: the sentence beginning with “However” requires a reference.

We thank the reviewer for picking this up. The passage has been revised with citation as suggested.

Figure 4: The legend text below the top level (group, num gene clusters) is blurry and hard to read.

We thank the reviewer for picking this up. The legend text has been revised as suggested.

Line 156-157 and line 233-234 is a misrepresentation of the reference. Ampicillin, vancomycin, metronidazole, and neomycin are not used together as surgical prophylaxis. Therefore, the statement “in line with most preoperative procedures” is not true. Surgical prophylaxis is most often accomplished with a single dose of cefazolin (Clinical Practice Guidelines for Antimicrobial Prophylaxis in Surgery). Indeed, there are few recommendations for multiple simultaneous agents; one of which being for complicated intra-abdominal procedures contaminated with fecal material. Please revise.

We thank the reviewer for these comments. The corresponding passages have been revised to highlight the varying combinations of antibiotics administered both in mouse models and in clinical procedures (lines 80-84 and 336-338).

REVIEWER COMMENTS

Reviewer #3 (Remarks to the Author):

Overall the manuscript is much improved. The authors have addressed most of my prior concerns. I have the remaining major concern and several minor comments

Major comment:

1. Figure 2: which taxa (fig. 2c) are contributing to the high abundance of ARGs at day 7 of treatment as shown in 2a? the authors comment that ARG enrichment is at 21 days, however there is no difference in ARG abundance between 7 and 21 days. It seems that the abundance of ARGs does not increase after initial treatment. There seems to be some discrepancy between Fig. 2a and 2c and 2d. Lines 115-118 are in direct contradiction to lines 144-146.

Minor comments:

2. The authors should provide a brief definition of selective sweep as that nomenclature may not be well known.

3. Supplementary Fig. 3A: several orange lines end in blue circles. Please revise. Please also perform repeated measures PERMANOVA and consider encircling the 95% confidence interval within groups to illustrate significant differences

4. Line 151-152: Figure 3a does not have a panel for the bacterial chromosome

5. Line 160-161: Figure 3a does not present the abundance of plasmids per se, but the relative abundance of AMR by category. Please revise.

6. Lines 182-184: Is the finding of Akkermansia involvement in HGT statistically significant?

7. Figure 4a: The text at the top fades out and cannot be completely read.

8. Line 230 should read "Clostridioides difficile"

9. Line 232-233: The combination of cefazolin, vancomycin, and gentamicin is not common practice for any surgery. The references cited recommend either cefazolin or vancomycin for most procedures. Please revise.

Response letter to the reviewers' comments

The detailed authors' responses to the reviewers' comments are listed in blue text below with the line numbers where modifications were made being highlighted in yellow. Additionally, the authors have numbered the comments (e.g. R1.1 for comment no.1 from reviewer 1) to improve readability.

Reviewer #3

Overall the manuscript is much improved. The authors have addressed most of my prior concerns. I have the remaining major concern and several minor comments.

We thank the reviewer for their acknowledgement of our revisions and the importance of our work. We greatly appreciate the reviewers' concerns and comments and have addressed these in detail in the responses to the comments below.

Major Comments:

Comment R3.1

Figure 2: which taxa (fig. 2c) are contributing to the high abundance of ARGs at day 7 of treatment as shown in 2a? the authors comment that ARG enrichment is at 21 days, however there is no difference in ARG abundance between 7 and 21 days. It seems that the abundance of ARGs does not increase after initial treatment. There seems to be some discrepancy between Fig. 2a and 2c and 2d. Lines 115-118 are in direct contradiction to lines 144-146.

Response: We acknowledge and thank the reviewer for this comment. We would like to clarify that Figure 2c was generated using high-quality (completion > 90% and contamination < 5%) metagenome-assembled genomes, i.e. MAGs. As a consequence of the antibiotic-treatment and the subsequent depletion of the microbiome, we were unable to recover any such MAGs at day 7 from the antibiotic-treated mice. This circumstance is reflected in Figure 2c. In order to clarify this point, we have updated the text in **lines 137-143** to reflect this.

With respect to the discrepancy described, we have updated the description in **lines 150-151** to indicate that the MAGs specifically from the Akkermansiaceae, Enterococcaceae and Lactobacillaceae families potentially acquired ARGs over time, rather than being encoded intrinsically.

Minor comments

R3.2: The authors should provide a brief definition of selective sweep as that nomenclature may not be well known.

Response: We thank the reviewer for this comment and have included a definition of the terminology in lines 47-48.

R3.3: Supplementary Fig. 3A: several orange lines end in blue circles. Please revise. Please also perform repeated measures PERMANOVA and consider encircling the 95% confidence interval within groups to illustrate significant differences

Response: As suggested by the reviewer, Supplementary Figure 3a has been revised, including 95% and 99% confidence intervals to demonstrate significant differences between the groups. The text and figure legend have also been revised in lines 103 and 546-548 respectively.

R3.4: Line 151-152: Figure 3a does not have a panel for the bacterial chromosome

Response: Figure 3a has been updated to include the bacterial chromosome. The respective figure legend has been updated in line 511.

R3.5: Line 160-161: Figure 3a does not present the abundance of plasmids per se, but the relative abundance of AMR by category. Please revise.

Response: Based on the reviewer's suggestions, the text in lines 166-168 has been updated.

R3.6: Lines 182-184 Is the finding of *Akkermansia* involvement in HGT statistically significant?

Response: We performed a Fisher's Exact test and did not find any statistical significance with respect to *Akkermansia*-mediated HGT. This is now reflected in the revised manuscript in lines 188-189.

R3.7: Figure 4a: The text at the top fades out and cannot be completely read.

Response: We thank the reviewer for this comment and the figure has been updated.

R3.8: Line 230 should read "Clostridioides difficile"

Response: Revised as suggested.

R3.9: Line 232-233: The combination of cefazolin, vancomycin, and gentamicin is not common practice for any surgery. The references cited recommend either cefazolin or vancomycin for most procedures. Please revise.

Response: We thank the reviewer for this comment and have revised the discussion in the revised manuscript in **lines 238-241**.

REVIEWERS' COMMENTS

Reviewer #3 (Remarks to the Author):

All comments adequately addressed. Well done!

Manuscript reference #NCOMMS-21-04264B

Response letter to the reviewers' comments

The detailed authors' responses to the reviewers' comments are listed in blue text below with the line numbers where modifications were made being highlighted in yellow. Additionally, the authors have numbered the comments (e.g. R1.1 for comment no.1 from reviewer 1) to improve readability.

Reviewer #3

All comments adequately addressed. Well done!

We thank the reviewer for their acknowledgement of our revisions and the critical feedback, which helped improve our manuscript.